# Variable Baseline and Flexible Configuration Stereo Vision Using Two Aerial Robots

**DOI:** 10.3390/s23031134

**Published:** 2023-01-18

**Authors:** Borwonpob Sumetheeprasit, Ricardo Rosales Martinez, Hannibal Paul, Robert Ladig, Kazuhiro Shimonomura

**Affiliations:** Department of Robotics, Ritsumeikan University, Kusatsu 5258577, Japan

**Keywords:** multirotor UAV, stereo vision, variable configuration stereo, aerial reconnaissance

## Abstract

In this work, a new method for aerial robot remote sensing using stereo vision is proposed. A variable baseline and flexible configuration stereo setup is achieved by separating the left camera and right camera on two separate quadrotor aerial robots. Monocular cameras, one on each aerial robot, are used as a stereo pair, allowing independent adjustment of the pose of the stereo pair. In contrast to conventional stereo vision where two cameras are fixed, having a flexible configuration system allows a large degree of independence in changing the configuration in accordance with various kinds of applications. Larger baselines can be used for stereo vision of farther away targets while using a vertical stereo configuration in tasks where there would be a loss of horizontal overlap caused by a lack of suitable horizontal configuration. Additionally, a method for the practical use of variable baseline stereo vision is introduced, combining multiple point clouds from multiple stereo baselines. Issues from using an inappropriate baseline, such as estimation error induced by insufficient baseline, and occlusions from using too large a baseline can be avoided with this solution.

## 1. Introduction

Aerial 3D reconstruction has become an important technology for various applications such as environmental research, urban planning, and disaster response. In the field of aerial remote sensing, numerous approaches have been proposed. Perhaps one of the most popular approaches is structure from motion (SFM) methods such as photogrammetry and, to some degree, simultaneous localization and mapping (SLAM).

For photogrammetry missions, careful mission planning and a flexible real-time mission management capacity are instrumental in achieving productive and safe acquisition missions [1]. The flight path of the deployed drone is planned to cover a large area of interest in order to create a large data set of overlapping aerial images for 3D reconstruction. This research aims to counter the necessity for extended flights and reduce the amount of movement the aerial robot needs to produce a 3D representation of an object of interest or area. Aerial robots, especially vertical take-off and landing (VTOL) multirotors’ flight time are severely limited due to their limited payload and hence, battery capacity. Active flight puts a larger strain on the battery than hovering in place.

Another approach of 3D reconstruction creation is visual simultaneous localization and mapping, or visual SLAM, such as the work proposed by [2]. While the purpose of SLAM also includes localization in addition to 3D reconstruction, a high amount of computation power is required. In order to create a comprehensible map, the amount of movement of the aerial robot is also fairly large. In the proposed method, by using a stereo vision approach as a foundation, the computation power is reduced by reducing the amount of data being processed in order to create a 3D reconstruction. The cost of computation for depth calculation is limited to only several selected stereo image pairs instead of processing every frame and depth data that would be acquired throughout the whole flight time when using a SLAM approach.

Light detection and ranging (LiDAR) is another popular device for 3D reconstruction [3]. The use of laser ranging devices results in extremely detailed 3D reconstruction. However, the limitation of LiDAR is their device cost [4]. The use of visible spectrum stereo vision only requires two RGB cameras as the mapping device. The combination of LiDAR and a camera has in the past years become available commercially. One of the most popular examples would be the use of such a system on smartphones or the commercial Realsense L515 LiDAR camera sensor. Moreover, the limitation of such systems is the operational range of the system: for example, the L515’s recommended range is up to nine meters. In this research, we introduce a 3D reconstruction method that only uses an RGB camera, a fundamental device that is already present on most of the aerial robots in such tasks, in combination with an additional motion-tracking capture camera on the ground that would not add to the drones’ payload.

In this paper, we introduce a 3D reconstruction system using two unmanned aerial vehicles (UAVs) utilizing the variable baseline and flexible configuration technique shown in Figure 1. The use of two UAVs facilitates a wide adjustment of the baseline as well as the possibility for a horizontal or vertical baseline configuration of the stereo cameras by adjusting the relative position of the UAVs. The idea of using a flexible baseline and multiple baseline stereo is quite a well-established topic in the field of computer vision. There are many researchers describing and implementing this idea such as in [5,6]. In this paper, a system of variable baseline and flexible configuration stereo is implemented on UAVs. By implementing the variable baseline stereo on UAVs, the adjustment of configuration becomes relatively easy and allows for a wide range of adjustability.

To our knowledge, there is not any research focusing on the idea of using a flexible stereo on VTOL UAVs. The closest to the proposed method is the work in [7], which focuses on the use of non-rigid stereo cameras on a fixed-wing UAV. Ref. [8] focuses on the use of multiple UAVs in cooperation with a ground robot for obstacle detection and path planning. In this paper, we focus on high-resolution map creation and the manipulation of the baseline as the main focus. Furthermore, we introduce a tracking method that could track the UAVs without the need to always maintain visual contact between the tracking device and the UAV.

## 2. Concept of Variable Baseline and Flexible Configuration Stereo Vision

In conventional stereo cameras, the left camera and right camera of the stereo pair are parallelly constrained. The relative position and orientation of the stereo pair is fixed; therefore, the characteristic and limitation is also limited to their setup. In contrast to conventional stereo cameras, the left and right camera of a variable baseline and flexible configuration stereo camera system are unconstrained. The two cameras of the stereo pair can move freely, allowing for the possibility of changing various settings of the stereo pair.

The prime advantage of being able to change the baseline is to be able to counter one of the principal problems of stereo vision, i.e., the depth estimation error that grows with respect to the target distance. This means that a large baseline can be used for farther objects to reduce the estimation error, while a smaller baseline can be used for closer objects to avoid occlusion from using excessive baseline.

As in [9], the estimation error of a stereo camera is defined by Equation (Equation 1). Here, $\epsilon_{z}$ is the depth estimation error in meters, *z* is the target distance in meters, *b* is the baseline distance in meters, *f* is the focal length in pixels, and $\epsilon_{d}$ is the disparity error in pixels.
(1)$$\epsilon_{z}=\frac{z^{2}}{bf}\cdot \epsilon_{d}$$

Figure 2 illustrates the equation by comparing the estimation error of stereo cameras with three different baseline distances: 0.5 m, 1.0 m, and 2.0 m. The interpretation of this graph is that for further objects, larger baselines are required in order to keep the error below a threshold. On the other hand, the use of an excessive baseline can also affect the quality of the disparity due to occlusion. It is almost impossible to predict the occlusion caused by an excessive baseline due to its dependence on many parameters, such as the size, shape, and position of the object in the area of interest. Therefore, it is extremely difficult to foresee the maximum baseline that can be used for a given object or environment. The proposed system is able to utilize the flexible baseline property of a flexible stereo system to adjust the baseline according to the appropriate corresponding target distance and characteristic of the area of interest.

In Section 4, an algorithm to fuse point clouds of various baselines into a more accurate resultant point cloud is introduced. Each baseline distance has its own effective range; therefore, in order to create a 3D point cloud of an area with a large variety of depth, a fusion of point clouds from multiple baselines is used to create a more accurate resultant point cloud than using only one baseline.

The second advantage of using a variable baseline and flexible configuration stereo is the ability to change the pose of the two cameras. Various stereo settings can be achieved. In this paper, vertical and horizontal stereo are utilized, as shown in Figure 3. Each setting has its own characteristic in terms of the resultant disparity image. Each setting is suitable for different kinds of targets, depending on various properties such as the dimension and shape of the target object, application of the resultant point cloud, etc. In Section 5, the advantage of the flexibility of the system in adjusting this setting in order to match the said properties is shown.

Although the flexibility and adaptability of a flexible configuration stereo opens the possibilities of having various settings, the most crucial challenge is the tracking of the pose of the cameras, i.e., the drones. For stereo vision, the accuracy of the depth estimation is closely dependent on the accuracy of the measurement of the two cameras’ relative poses. Since the drones can move freely and at any distance from each other, it can be challenging to keep track of the relative pose accurately. Furthermore, in addition to knowing the relative pose, the absolute position of the drone is also crucial for projection of the point cloud to the world coordinate. Therefore, in Section 3, the implementation of the tracking method using a combination of a marker-based motion-tracking device and the drones’ built-in tracking sensors is detailed.

## 3. Tracking System of the Two UAVs

### 3.1. Materials

In this research, two multirotors are used as reconnaissance agents. Two DJI Tello micro UAVs are used as the experiment platform due to their ease of setup and their compact size. With the diagonal length of 0.11 m, the distance between the camera, positioned in the center, to the edge of the frame is relatively small. Thus, the smallest baseline that can be achieved without the risk of the drones crashing or interfering with each other’s thrust is about 0.4 m. Each DJI Tello is equipped with a front-facing monocular camera with the properties described in Table 1. Each drone’s camera is used as one camera of a stereo pair; by combining both drones’ cameras, a stereo image pair is formed.

Tracking accuracy is dependent on the accuracy of the depth estimation. Thus, a portable motion capture device, the Optitrack V120 Duo shown in Figure 4, is used as the primary tracking method in this research. The device is a dual-infrared motion capture (mocap) camera that does not require a ground plane setup or cameras’ position calibration such as wanding or ground-plane calibration prior to use in comparison to normal mocap systems. Furthermore, the compact size allows the device to be mounted on a ground vehicle or a ground-based station. The mocap device tracks the infrared illuminator markers, which are small reflective spheres that are put on the drones. A 6-DOF pose, including the position and orientation of the drones, is measured up to the distance of 4 m away from the mocap device. Additional information of the mocap characteristic is shown in Table 1.

### 3.2. Tracking Method

Figure 5 shows the notations used in the tracking scheme. The tracking of the two drones, {D1} and {D2}, is primarily completed by the motion capture device {M}. The 6-DOF pose of the drones relative to the mocap device, i.e., translation and rotation from {M} to {D1} and {D2} is obtained via mocap tracking. Moreover, the translation and rotation of the mocap device to the world coordinates {W} can be obtained by comparing the mocap measurement with the internal odometry of each drone. Using this coordinate scheme, the pose of the drones {D1}, {D2} relative to a static world coordinate frame {W} is known. The use of this information will be crucial for point cloud fusion, which will be described later in Section 4.

However, there is an issue in this tracking scheme if only the mocap device would be used for tracking. The mocap device has a limited field of view. Once one or both of the drones leaves this field of view, the tracking for this drone would be lost. Therefore, a tracking scheme where the drones’ odometry is used when the drones are out of the field of view of the mocap device is implemented and described in Section 3.

An Extended Kalman Filter (EKF) is used for localization of the drones by fusing the odometry data with mocap tracking data. As shown in Figure 6, when the drones are within the field of view of the mocap, the drones are tracked solely using the pinpoint accurate pose data from the mocap device. When the drone leaves the field of view of the mocap, the last known pose detected by the mocap will be registered, and the EKF switches to use the odometry of the drones in combination with IMUs to track the drones. In this fashion, the drones can be tracked outside the field of view of the mocap within the following two requirements. Each drone needs to pass through the field of view of the mocap at least once to initialize the pose offset between the odometry and mocap. Since the drones’ odometry is not very precise and tends to drift over time, the drones need to come back into view every once in a while to adjust the drift after the drones move a significant distance, i.e., after using large baselines.

Another advantage of using the drones’ odometry is to use the drones’ orientation information to adjust the ground plane calibration. Since the drones’ orientation obtained via a flight controller is always the orientation with respect to the ground in the world coordinates, the IMU’s pitch and roll values are always relative to the orientation of the world ground plane. This information is used to inversely calculate the orientation of the mocap device itself by comparing the orientation of the drones relative to the mocap device with the orientation of the drones measured by an IMU. Furthermore, since heading data are obtained from the magnetometer of the IMU, it is possible to obtain the heading of the drones relative to the world coordinates as well. Consequently, it is therefore possible to calculate the heading of the mocap device itself by averaging the heading of the two drones with respect to the mocap device.

As shown in following Equation (Equation 2), two interpretations of mocap orientation can be calculated using either the rotation matrix of the primary drone’s IMU relative to the world frame $R_{D1}^{W}$ or the rotation matrix of the secondary drone’s IMU relative to the world frame $R_{D2}^{W}$.
(2)$$\begin{array}{c} R_{M}^{W}=R_{D1}^{W}\;{R_{D1}^{M}}^{\prime }\\ R_{M}^{W}=R_{D2}^{W}\;{R_{D2}^{M}}^{\prime } \end{array}$$

These rotations from the IMU are multiplied by the inverse of the rotation matrix of the primary drone’s IMU and the secondary drone’s IMU relative to the mocap frame, $R_{D1}^{M}$$R_{D2}^{M}$. The final rotation matrix from the motion capture frame {M} to the world frame {W}, denoted as $R_{M}^{W}$, can be calculated by averaging the calculated two values in Equation (Equation 2). The averaging is completed by converting the rotation matrices into quaternions. After averaging by using the algorithm described in [10], the resulting quaternion is converted back into a rotation matrix for later use in extrinsic parameters calculation. Using the described method, additional sensors such as an additional IMU on the ground side is not needed for calibration of the ground plane orientation. This operation needs to be completed only once at the beginning, before tracking begins, similar to the ground plane calibration process that is completed before using the mocap camera.

### 3.3. Relative Pose Calculation

A crucial information that the tracking obtains is the relative pose of the two drones. This information is used in the image processing and point cloud processing step described in Section 4. The accuracy of the system is significantly dependent on the accuracy of the relative pose measurement. In the system, by having the mocap device as the reference point of tracking, it is possible to calculate the relative pose between two drones. The relative rotation between the first and the second drone can be calculated using,
(3)$$R_{D2}^{D1}=R_{D2}^{M}\;{R_{D1}^{M}}^{\prime }$$
with all variables with respect to Frame *M*, which is the mocap frame. $R_{D1}^{M}$ and $R_{D2}^{M}$ are the orientation of the drones measured by the mocap device; therefore, the orientations are with respect to the mocap frame *M*.

After obtaining the relative orientation, the relative position can be calculated with,
(4)$$T_{D2}^{D1}=T_{D2}^{M}-R_{D2}^{D1}\;T_{D1}^{M}$$
where $T_{D2}^{D1}$ denotes the relative position from the secondary drone to primary drone, $T_{D1}^{M}$ and $T_{D2}^{M}$ denotes the position of the first and second drone measured by mocap. The equation would result in translation from the first drone to the second drone.

Using the above calculation, the relative attitude of the two drones using the mocap device is acquired. However, the posture measurement of the mocap is only available if both of the drones are present inside the field of view of the mocap. When one or both of the drones leaves the mocap, the drones’ odometry as well as their orientation sensors must be used instead for the posture acquirement. Therefore, we introduce a tracking method that is used in the said situation. To calculate relative rotation, the orientation of the drones relative to the ground plane obtained from the drones’ IMU is used, and the calculation is shown in the following Equation (Equation 5).
(5)$$R_{D2}^{D1}=R_{D2}^{W}\;{R_{D1}^{W}}^{\prime }$$

As explained above, the orientation of the IMU is always relative to a ground plane, and the heading is relative to the compass. Therefore, another relative rotation can be calculated by using the same calculation as in Equation (Equation 3) with rotation measured by the IMU. In Equation (Equation 5), all rotation matrices are with respect to the ground plane and ground heading.

Acquisition of the relative position requires more effort, since each drone’s odometry has a different coordinate reference point, depending on its takeoff point. Therefore, the mocap frame *M* is used as the reference point to calculate the total translation between the coordinate frames of both drones’ internal tracking. Each of the drones need to fly into the field of view of the mocap device at least once in order to synchronize the offset between the coordinate of the drones’ odometry and the mocap.

$T_{O1}$ and $T_{O2}$ indicate the position offsets of D1 and D2 described in Figure 6, which are acquired using the following Equation (Equation 6).
(6)$$\begin{array}{c} T_{O1}=T_{D1}^{M}-R_{M}^{W}\;T_{D1}^{W}\\ T_{O2}=T_{D2}^{M}-R_{M}^{W}\;T_{D2}^{W} \end{array}$$

These offsets translate the coordinate system of the drones’ odometry to mocap. Finally, total relative translation from the first drone to the second drone using odometry can be calculated using the following Equation (Equation 7) where $T_{O1}$ and $T_{O2}$ are the latest known offset.
(7)$$T_{D2}^{D1}=T_{D2}^{W}-T_{O2}-R_{M}^{W}\;R_{D2}^{D1}\;\left(T_{D1}^{W}-T_{O1}\right)$$

Notice that in the system, there exist two relative rotations and positions of the drones from Equations (Equation 3) and (Equation 4) acquired from mocap and Equations (Equation 5) and (Equation 7) acquired from the combination of IMU, the flight controller’s odometry and mocap. The relative pose acquired solely from the mocap would only be present if both of the drones are in the field of view of the mocap. Once any of the drones leaves the field of view, the system would switch to use the relative pose acquired from the IMU instead. By using this method, the drones movement would not be limited to just inside the field of view of the mocap but could also reliably operate outside of the mocap’s FOV. This results in the drones being able to move anywhere and form any desired baseline configuration as long as the drift in the odometry is not too large such that it affects the ability to correctly rectify the image or exceedingly affect the accuracy of the depth calculation. In this proposed system, the positional accuracy has been measured by comparing the odometry measurement with motion capture measurement. The result of the average drift occurs at roughly 0.024 m for each meter that the drone moves. Even then, the drone can come back into the field of view to adjust the drift and once again leave the field of view for further operation.

## 4. Image Processing and Point Cloud Processing

Parameters needed for image processing and the relative pose of the two cameras are acquired from the previous section, camera intrinsics are acquired using the camera calibration method described in [11], and the raw images are acquired from the front monocular camera from each drone. The first drone’s bearing the D1 coordinate frame will be termed the “primary drone”, and the second drone bearing the D2 coordinate frame is termed the “secondary drone”. The primary drone is the left drone in the horizontal case, and it is the lower drone in the vertical case. In this section, the image processing process from raw images to the resultant point cloud is explained. Furthermore, a multiple baseline fusion algorithm that fuses multiple point clouds from multiple baselines into a high-resolution point cloud is introduced.

### 4.1. Stereo Image Processing

Overview of the image processing steps is shown in Figure 7. The first step of depth image processing is to acquire stereo correspondence and their corresponding disparity from the stereo image. The images are calibrated and rectified according to the intrinsic parameters including camera matrices $K_{L}$, $K_{R}$ and distortion matrices $D_{L}$, $D_{R}$. The rectification matrices of the images are acquired using the relative pose of the two drones by computation of the algorithm described in [12]. Projection matrices $P_{L}$, $P_{R}$ and rectification matrices $R_{L}$, $R_{R}$ are obtained. The images are then rectified to a parallel stereo image pair using the obtained rectification matrices.

After obtaining the parallel stereo pair, the Quasi-Dense stereo-matching algorithm described in [13] is used for stereo matching and correspondence. This particular method was chosen, since the measurement of the pose between two drones in the proposed method is not always without errors. Practically, the measurements of the drones pose are affected by various factors, such as measurement delay, measurement noises, and odometry drifts. Rectification of the stereo image pair is hardly perfect as well. There always exists the slight measurement error that would cause the rectified images to be slightly shifted or misaligned. This research is focused on emphasizing the use of a large baseline stereo. Slight measurement errors of the relative orientation would mean a large error in the rectification process. The Quasi-Dense stereo-matching algorithm provides excellent robustness for an imperfectly rectified stereo image pair and hence was chosen in this research. The performance comparison of three stereo matching algorithm is shown in Figure 8.

### 4.2. Point Cloud Processing

After obtaining the disparity image, the next step is to reproject the disparity value into the 3D space. Using [12], the reprojection matrix *Q* is also obtained. This reprojection matrix is used for the reprojection of disparity images into a 3D space, i.e., a point cloud. Disparity values obtained from the previous step are values of disparity in pixels of each image point [u,v] of the primary drone’s camera (left camera in horizontal stereo setup, lower camera in vertical setup). Therefore, the resultant point cloud from the reprojection would be with respect to the primary drone’s coordinate frame, D1.

In this research, the primary drone’s camera is on a hovering drone; therefore, the camera is always moving. By knowing the primary drone’s position relative to the world coordinate, transformation of the resultant point cloud from the coordinate frame of the primary drone D1 to the world coordinate frame *W* is possible.

### 4.3. Multiple Baseline Fusion Algorithm

An algorithm for baseline fusion is introduced. By merging point clouds from multiple baselines, it is possible to create a resultant point cloud with a controllable estimation error. As explained in the introduction, each baseline has their own effective range when a depth estimation error constraint is introduced. Using this algorithm, a large baseline can be used for long range depth estimation while using a smaller baseline to cover for occlusion caused by an excessive baseline.

In this research, the two cameras are unconstrained, therefore allowing multiple configurations and baselines. To maximize the advantage of this property, in this algorithm, four properties can be defined by the operator:$n_{B}$: The number of baselines that the user desired.$\epsilon_{c}$: The error threshold that the user needs in the resultant point cloud.$z_{max}$: The maximum distance that the $\epsilon_{c}$ should hold.(optional) $z_{min}$: The minimum distance of the object of interest.

The first step is to find the largest baseline that would fit the given constraints $\epsilon_{c}$ and $Z_{max}$. Using the depth estimation error in Equation (Equation 1), while assuming the disparity error $\epsilon_{D}$ to always be 1, the largest baseline can be determined by the following equation.
(8)$$b_{max}=\frac{z_{max}^{2}}{\epsilon_{c}f}\epsilon_{d}$$

After obtaining the largest baseline, the operator can first check if such a baseline with such a constraint would be physically possible, depending on the system set up and the drones’ camera characteristic such as field of view and focal length. For example, in this case, a small drone is used; therefore, there is a height and range limitation at about 20 m due to the limitation of the WiFi communication device used to communicate with the drones. Additionally, if the baseline is too large in the case where the drones’ camera has a limited field of view, the overlap of both cameras’ fieldss of view would be too small to include sufficient useful information. If $b_{max}$ is usable, it is able to proceed to the next step to obtain which baseline is to be used. Same goes for the smallest baseline, which would be limited by the size and thrust interference of the specific drones.

Based on $n_{B}$, the corresponding number of baselines is derived by dividing the range between the minimum value of the z axis $z_{min}$ and its maximum value $z_{max}$ into smaller segments $n_{B}$. Then, the baseline distance that would meet the error threshold in each corresponding segment is derived through Equation (Equation 8). Each section of estimated distance represents the usable range of each derived baseline.

For example, we define an example case using the proposed method for better scaled visualization and understanding. In this example case, the conditions are defined as follows: $n_{B}$ = 3, $\epsilon_{c}$ = constant 0.5 m, $z_{max}$ = 40 m, $z_{min}$ = 0 m. The derived baselines using the algorithms are 0.39 m, 1.58 m, and 3.55 m. Point clouds using the derived three baselines would then be acquired using the drones. After that, all point clouds are combined into a resultant point cloud by trimming each point cloud at their respective range based on each segments; only depth data in the respective section of each baseline are retrieved. Therefore, as shown in Figure 9, the first baseline of 0.39 m is used for object distances between 0 and 13.33 m, the second baseline of 1.58 m is used for objects of distance between 13.33 and 26.66 m, and the final baseline of 1.43 m is used for objects of distance from 26.66 m onwards.

However, in many real-world cases, the predicted baselines are not always perfect. As mentioned in Section 2, the quality of the disparity map created by each baseline is also dependent on many unpredictable constraints, such as the size and shape of the object in the area of interest. In such cases, the baseline distances are chosen after the stereo images have already been taken and the quality of disparity images has already been verified. The proposed setup, where two drones are used, allows the operator to preview the disparity image at each baseline first, verify their quality, and adjust the baseline accordingly if the disparity is of low quality. It also allows for the ability to record a linear set of baselines in a collection of pictures taken continuously in a single drone movement. In such cases, the baseline distance is not derived through the equation but fixed to practical constraints. The point of trimming of each baseline can be obtained using the following equation,
(9)$$z_{trim}=\sqrt{\frac{\epsilon_{c}bf}{\epsilon_{d}}}$$
which is derived from the earlier Equation (Equation 8) where $b_{max}$ is substituted with the desired baseline *b*.

After finding the trimming point of each baseline, the point clouds obtained from stereo images at each baseline are combined together using the trimming points. The resultant point cloud map would have the estimation error under the error constraint given by the operator. The only disadvantage is the loss of image–point to object–point correlation property that exists in normal point clouds generated from a single stereo image pair. However, in this research’s focused application of 3D reconstruction, such a property is not necessarily required.

## 5. Experiment Results

In this research, the general characteristic of the variable baseline stereo configuration is explored by experimenting with various stereo setups in the simulation software AirSim [14]. The characteristic of the camera in AirSim, including focal length and field of view, is set to a similar value as the camera of the DJI Tello drones. After confirming the general characteristics of the system, a real-world experiment is conducted to verify the concept.

### 5.1. Horizontal and Vertical Setup Comparison Experiment

In the first experiment, the characteristics of horizontal stereo and vertical stereo are compared. Stereo image pairs are taken in two modes, horizontal and vertical stereo as shown in Figure 10. From each mode, 60 stereo image pairs of baseline 0.05 to 3.0 m are taken using the simulation software AirSim. All the images are shifted on a single axis by the steps of 0.05 m; therefore, all images are perfectly parallel. The images are then undistorted, rectified, and stereo matched. The resultant disparity image and depth estimation of both the horizontal and vertical stereo system are compared at each corresponding baseline.

In Figure 11, the disparity image of both setups at the baseline of 1.0 m is compared. The first noticeable difference is the loss of information due to the non-overlap area caused by the field of view. The loss occurs on the left side of the disparity image in a horizontal setup, while the loss occurs on the lower side of the image in a vertical setup. Depending on the purpose of the result, each setup can be chosen to match the requirement. In the case of this research, the point clouds generated from the disparity images are used for the baseline fusion. Using the algorithm explained in the last chapter, closer points are trimmed off from the point cloud of larger baselines. Accordingly, most of the time, closer objects are located at the lower part of the disparity image. The missing information would reduce the depth computation time of the information of these closer points in a large baseline, which would eventually be trimmed by the baseline fusion. At the same time, these closer points would be properly covered by a point cloud of smaller baselines, which has a lower chance of being trimmed by the algorithm. Furthermore, in the horizontal setup, the loss of horizontal disparity grows larger as the baseline increases, resulting in each point cloud having a gradually reduced horizontal field of view. When the point clouds are fused, the data on the left side would be missing, especially on farther objects where larger baselines are used. Therefore, a vertical setup is more suited in general for the task focused in this research.

The second difference is the occlusion caused by camera shift. In a horizontal setup, the occlusion occurs behind vertical surfaces, such as a wall, or the vertical side of the windows frame. On the other hand, the occlusion of a vertical stereo system occurs on horizontal surfaces, such as the gable of the house. When the system is used on a different type, shape and size of objects in the area of interest, either setup can be chosen in order to extend the reach of depth information in the stereo pair. For instance, when used for mapping a tall building, such as a tower or a lighthouse, either setup can be used if the accuracy of the building itself is the only concern. However, in a horizontal setup, the occlusion caused by the building would occur in a large area surrounding the sides of the building. This means that more information on the area behind the building is lost due to occlusion. On the other hand, when a vertical setup is used, the occlusion at the sides of the building can be minimized. Therefore, the ability to swap between the two setups is beneficial for choosing the right setup for each environment.

The third property is the effects on the depth estimation of the point cloud. The depth estimation of both setups is compared. Figure 12 shows their corresponding point cloud seen from a top–down view. There are little to no major differences at a glance. The two depth estimations were compared mathematically. The root mean squared difference of the depth estimation from both setups is calculated across all baselines. Only matched points are taken into account. The average RMS difference of all baselines for both setups is 0.021 m, meaning that there are no major differences in depth estimation considering the scale of the difference compared with the mean depth estimated in this experiment, which is about 30 m.

The fourth difference is the possibility of the drones forming each setup. As mentioned in Section 3, there is a physical limit to the closest distance that the drones can hover next to each other. In the horizontal case, it is limited to the size of the drones; in this system, the limit is about 0.3 m. On the other hand, the limit is generally farther in the vertical case, which is due to the limit being dictated by the thrust of the drones. Depending on the type of the drone used, the closest distance that the drones can fly on top of each other varies. In the system proposed in this paper, according to the test, the limit is around 0.4 m.

### 5.2. Disparity Distribution Analysis Experiment

In this experiment, the distribution of depth estimated from the stereo disparity image is analyzed in order to find the relationship between the theoretical estimation error value and estimation error in practice. Similar to the earlier experiment, 120 perfectly aligned stereo image pairs were taken from AirSim from two different locations, each with 60 pairs and of various baselines.

From the benchmark area of interest shown in Figure 13, the depth estimation distribution is analyzed. Some of the selected distribution can be seen in Figure 14 shown with each respective baseline distance. In such a perfectly parallel cases, the gap between each populated depth can be clearly seen. The width of the gap is roughly equal to the theoretical estimation error given by Equation (Equation 1). This means that the actual error estimated by the system will fall in the range of the theoretical estimation error.

In Figure 15, the estimated depth from the system is compared with the ground truth obtained from the simulation software. The estimation error falls in between the theoretical values. Therefore, using these benchmarks as a basis, it is valid to assume that the trimming point given by the baseline fusion algorithm will hold in real-world applications as well.

### 5.3. Baseline Combination in Simulation Experiment

In this experiment, the baseline combination algorithm is used to create a resultant point cloud. The same area of interest as in Figure 13a is used. Vertical configuration stereo images of an area of interest are taken according to a set of set conditions. The set conditions are $n_{B}$ = 3, $\epsilon_{c}$ = constant 0.5 m, $z_{max}$ = 40 m. The additional cameras and stereo-matching characteristic parameters are *f* = 900 pixels and $\epsilon_{D}$ = 1 pixel. In this experiment, an additional customization property, $z_{min}$ = 10 m, indicating the minimum distance of interest, is added to the condition. This additional property changes the way of depth segmentation, which normally starts at 0 to $z_{max}$; in this case, only the depth of range $z_{min}$ to $z_{max}$ is considered. $z_{min}$ can be added to cases where the resolution of the resultant point cloud at a specific range is prioritized: for example, in cases where the rough depth of a specific high-priority region in an area of interest is somewhat known. In this case, the house is the focus priority. The rough length of the front of the house measured using ground truth is around 35 m. The depth of the front yard to the farthest wall is estimated to be somewhere around 10 to 40 m. Therefore, the segments are divided into three segments: 10 to 20 m, 20 to 30 m, and 30 to 40 m.

The estimated baseline for each segment is 0.88 m, 2.0 m, and 3.5 m. For simplicity, the baselines are rounded to integers of equal intervals: 1, 2 and 3 m. The point cloud of each baseline is shown in Figure 16. At this point, the rough estimated depth of the house is known by looking at the point cloud, which is roughly between 20 and 40 m, which is a bit off from the initial guess. In practice, if the rough estimate is far off from the initial guess, it is still possible to use the algorithm to recalculate by changing the $z_{min}$ and $z_{max}$ properties, calculate new baselines, and retake the point cloud once again. However, in this case, the initial estimate is within the expected range; therefore, we proceed to the next step.

The point cloud from each respective baseline and segment is retrieved, while the rest are trimmed. The retrieved points are combined into a resultant point cloud shown in Figure 17. The resultant point cloud is compared to the initial point cloud of each baseline. In Figure 16, (a) is the point cloud of a 1 m baseline. It is noticeable that the resolution of the point cloud in the farther distances is extremely low due to gaps in depth estimation, i.e., the estimation error. On the other hand, (b) and (c) which are made by larger baselines, have better resolution. However, on (c), some of the details closer than 20 m cannot be matched due to occlusions caused by an excessive baseline, and therefore, no points are visible in the point cloud. By combining the closer points from (a) and higher-resolution points from (b) and (c), a more complete resultant point cloud is created.

### 5.4. Real-World 3D Reconstruction Experiment

In the final experiment, the system is tested in a real-world application. Two drones are used to retrieve stereo images, while the tracking method described in the earlier section is used. The mocap device, Optitrack V120 Duo, is placed on the ground at an unknown angle and heading. The experiment area with similar conditions to the second experiment is chosen. As seen in Figure 18, the area has objects in various ranges, where the farthest trees at the end of the parking lot are roughly 70 m from the origin point of the primary drones camera. The focus priority is the white warehouse on the left side, which has its range measured by a laser range finder at around 30 m. Due to the nature of the area of interest, the details of the building on the left-hand side should be visible in the resultant point cloud that would otherwise be lost in the horizontal setup, so the vertical setup is chosen in this experiment.

First, the ground plane and heading calibration is initiated. The roll, pitch, and heading of the mocap device is measured as 2.57 deg, 68.85 deg, and 81.21 deg heading, indicating that the mocap camera is pointing up 67 deg in order to see the drones as high as possible before leaving the field of view. Using the same constraints as in the earlier experiment, the same baseline distance of 1, 2 and 3 m is used.

Two drones are used to acquire stereo image pairs. The primary drone hovers at a point where the area of interest is clearly seen in its camera. The secondary drone is controlled autonomously to each baseline distance, and in this case, it is controlled to hold at altitudes of each baseline for 5 s and proceed to the next. An image buffer at a control station records the video from the drones camera and position as the drones move. After that, the most parallel image pairs from each baseline distance are chosen and processed into the disparity image. The stereo image pair and their corresponding disparity image can be seen in Figure 19. Three disparity images were taken; however, as seen in Figure 19, objects beyond 30 m, or beyond the white warehouse, are barely visible at the 3 m baseline due to the nature of the scene and the rectification, even though the 3 m baseline’s segment is supposed to be used for distances of 30 to 40 m. Furthermore, the rest of the objects are too close, and therefore, mostly noises are present in the disparity image. In this stage, the disparity image of the 3 m baseline is deemed unnecessary to add to the baseline fusion. Only the 1 and 2 m baselines are used.

The point cloud projection from both baselines is seen in Figure 20. The low resolution of the white warehouse in the 1 m baseline due to high estimation error can be clearly seen. On the other hand, the lack of road on the closer distance can be seen in the 2 m baseline. Therefore, both baselines are fused using a trimming point at 20 m.

The resultant point cloud can be seen from Figure 21. The resultant point cloud is compared to the ground truth information obtained from Google Maps [15]. In Figure 22, the shape of the key points in the point cloud is matched by stretching the point cloud to fit the red lines, which are drawn based on their corresponding positions in ground truth. The scale of the point cloud can be seen as a little bit smaller than the ground truth by comparing the 20 m scale between the measured point cloud showing in a yellow arrow and ground truth showing in a white ruler line. The positions of the objects are mostly correct, with some imperfection at the building on the right side, which seems to be curved toward the center at a farther range due to imperfection in rectification.

## 6. Discussion

The result of the simulation experiments shows the characteristic of the system, which is used as the basis for conducting the real-world experiment. The results of the first experiment show the characteristics of both horizontal and vertical stereo setups, which have their own advantages and disadvantages. The appropriate setup can be chosen correctly based on these characteristics, as seen in the real-world experiment where a vertical setup is used due to the nature of the area of interest. In addition to choosing only one setup, the further improvement would be to be able to fuse both vertical and horizontal stereos using the algorithm proposed in [16]. By increasing the number of mapping agents, in this case, the number of drones, multiple views including the combination of both horizontal and vertical setups can be used. The fusion of the two setups would eliminate the occlusion described in the experiment in Section 5. Another possibility of the combination of both horizontal and vertical setups is the possibility to refine stereo-matching algorithms by using multiple drones. Similar to the work proposed in [17], the accuracy of disparity calculation can be increased beyond one pixel, therefore achieving subpixel accuracy. This would allow an increase in the depth estimation accuracy without the need to increase the resolution of the cameras.

The second experiment proves the principal characteristics of stereo vision where the error of depth estimation increases quadratically to the estimation distance. By knowing the characteristics of the system, the simulation experiment leads to the baseline fusion algorithm as shown in the third experiment. Through the use of multiple baselines and the fusion algorithm, a high-resolution resultant point cloud is created.

The real-world experiment shows that the system can be used in the real-world application. Still, the limitation of the system is still present in the tracking system. The tracking method using a motion capture device has its limitations because of its dependency on infrared visibility. In a small separate experiment, the usability of the motion capture device in an outdoor environment is tested. During day time with strong sunlight, the visibility of the motion capture markers is interfered by the infrared light from the reflection of sunlight. Therefore, in the real-world experiment, we chose the environment behind a building where there is shade. The experiment was carried out during the evening where there is less sunlight.

For that reason, a better alternative to the motion capture system is considered in the future work. In our previous research, we used the vision-based fiducial markers described in [18] for drones tracking, however, due to performance problems and a problem with camera limitations, mocap is chosen as an alternative. If the issues are resolved, such markers can be used again in place of mocap. Other choices considered are inside–out tracking methods such as dual SLAM introduced in [19] and the collaborative localization of stereo cameras on UAVs described in [20].

## 7. Conclusions

In this paper, we introduce a 3D reconstruction system based on multiple UAVs’ stereo vision. Using a variable baseline and flexible configuration technique, multiple baselines can be used and horizontal or vertical stereo configuration can be chosen according to the characteristic of the region of interest. Several point clouds are acquired using several baseline distances. A point cloud fusion algorithm is introduced to create a more comprehensible and more accurate point cloud from the fusion of point clouds from several baselines. Experiments show the characteristics and feasibility of the system in simulations. Moreover, real-world implementation of the system is introduced as well as the combination of marker-based tracking and internal odometry for relative pose tracking. The experimental results show the application in a real-world implementation using the proposed system. The system shows a novel method of 3D reconstruction that requires minimal active flight distance of the UAVs to reconstruct a large area of interest.

Compared to existing methods, the advantages and disadvantages of our method can be summarized as follows. The advantage of our method over other methods mentioned in Section 1 is the ratio between active movement distance of the drone over the area covered in 3D reconstruction. The experiment result has shown the use of only 3 m of active flight of one drone and static hovering of another drone to create a depth reconstruction of an area of 620 m${}^{2}$ with 1.25% depth estimation error at 40 m distance. However, by limiting the flight distance, the disadvantage of our method is the limitation of the shape of the reconstruction area. The shape of the reconstruction area using the proposed method is fixed to the shape of an intersection area of the two cameras’ FOV, which in the case of the front-facing camera would be the shape of a triangle when seen from above. The next difference is the occlusion; since two front-facing cameras are used and only one reference point is used for all the baselines, the occlusion behind objects is unavoidable. For that reason, the proposed system is more fitted in the task of reconstructing an open area. Still, a large object such as buildings or bridges can still be reconstructed as long as the occlusion of the object on the area behind it is tolerable. Moreover, the use of more than two agents, such as by increasing the number of drones and thus increasing the point of view of the system, can potentially reduce or eliminate the occlusion problem. The final advantage of our system over the classic SFM approaches is the ability to maintain the 3D monitoring of an area in close-to-real time when the two drones are kept hovering at a constant baseline. In terms of computation time compared to conventional fixed stereo cameras, the proposed system has additional computational tasks, namely the relative posture calculation and implementation of the EKF. These additional tasks, compared to the computation time taken by the stereo-matching algorithm, take a relatively small amount of computation time. The system can be used as a very large stereo camera with little additional computation power required. Consequently, 3D monitoring of the area is possible, which enables further application such as the monitoring and guidance of another working agent, such as a ground robot or another manipulation drone that would operate in the area.

## Figures and Tables

**Figure 1 sensors-23-01134-f001:**
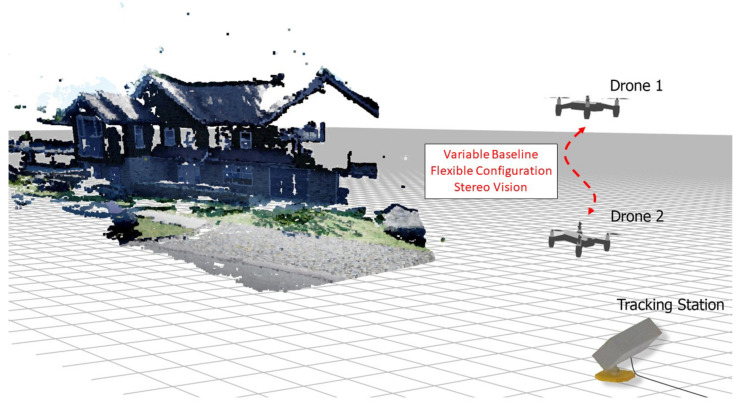
Graphical render of proposed system, where two drones are used as the reconnaissance agents for 3D map creation. A ground-tracking device is used for tracking.

**Figure 2 sensors-23-01134-f002:**
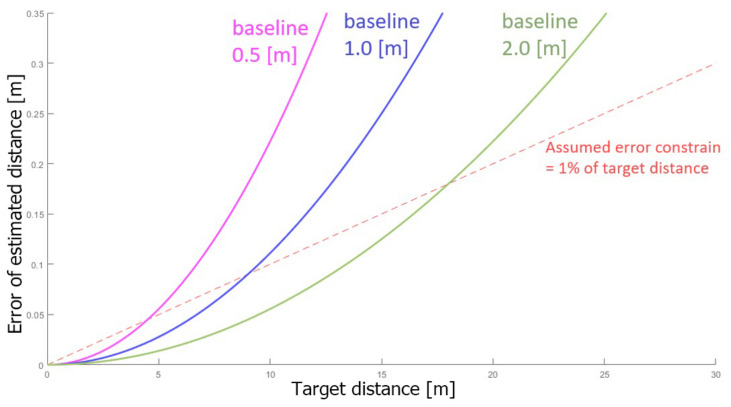
The theoretical value of estimation error on y-axis vs. target distance on x-axis. The graph shows the estimation error of three different baselines shown in three solid color lines. For reference, the virtual error constrain of 1% of the target distance is shown in a dashed red line.

**Figure 3 sensors-23-01134-f003:**
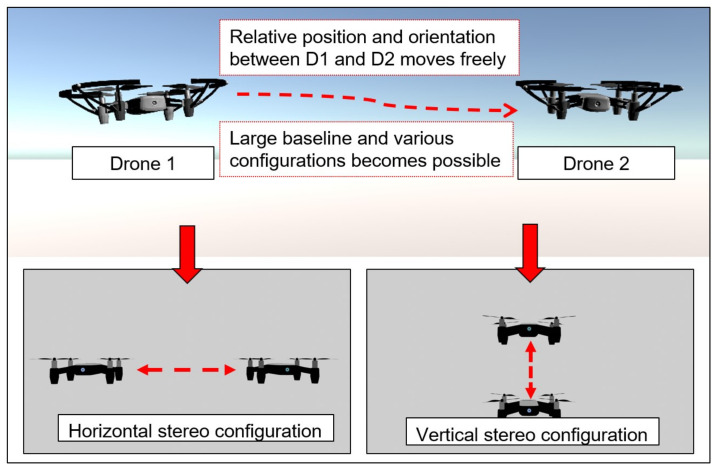
The possibility of changing multiple setups of the flexible configuration stereo.

**Figure 4 sensors-23-01134-f004:**
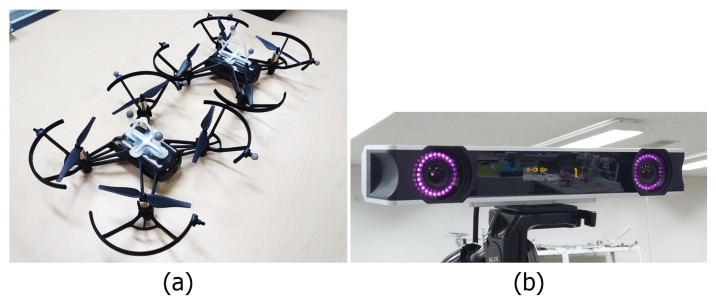
(**a**) DJI Tello micro UAVs with motion capture markers attached. (**b**) Portable dual-lenses motion capture device Optitrack V120 Duo.

**Figure 5 sensors-23-01134-f005:**
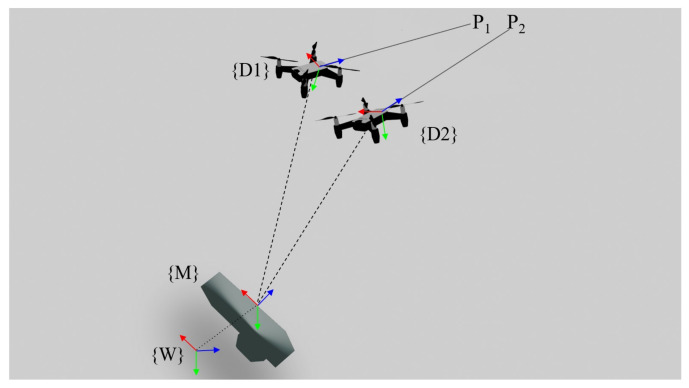
Notation of translations and rotations in the system.

**Figure 6 sensors-23-01134-f006:**
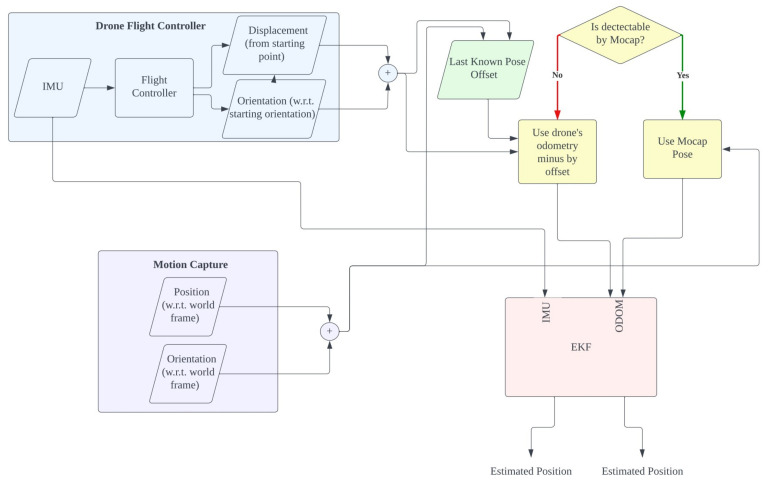
Flowchart of the tracking scheme using the combination of motion capture and odometry data.

**Figure 7 sensors-23-01134-f007:**
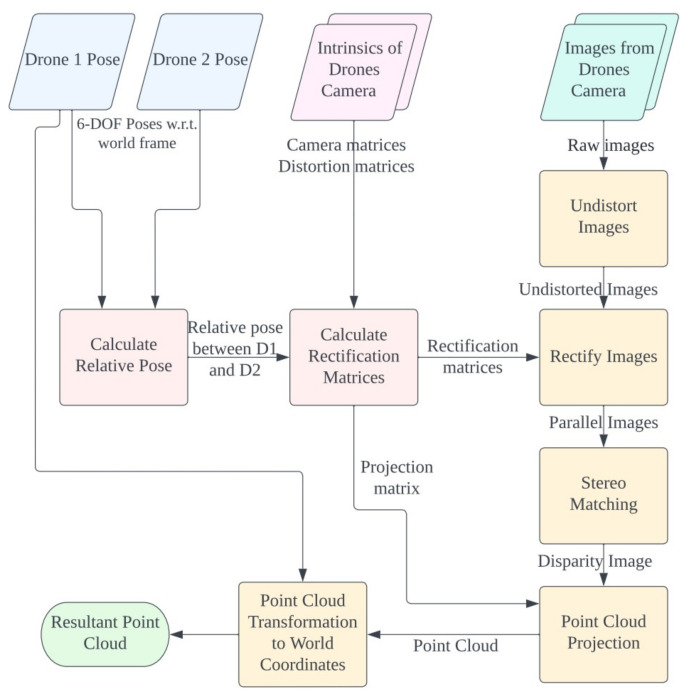
Flowchart of the process in image processing along with data used to process from raw images to 3D point cloud.

**Figure 8 sensors-23-01134-f008:**
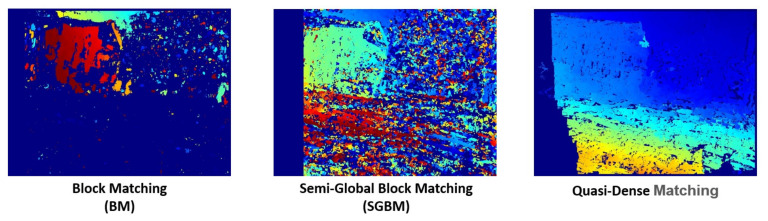
Comparison of Block Matching (BM), Semi-Global Block Matching (SGBM), and Quasi-Dense matching algorithm in a sample non-perfectly rectified image pair.

**Figure 9 sensors-23-01134-f009:**
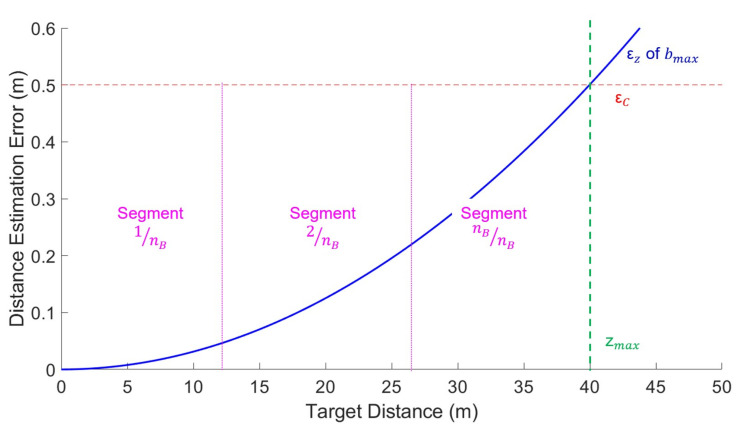
Graph of an example case of baseline fusion. x-axis is distance, while y-axis is depth estimation error in meters. This example case is the case where the given conditions are $n_{B}$ = 3, $\epsilon_{c}$ = constant 0.5 m, $z_{max}$ = 40 m.

**Figure 10 sensors-23-01134-f010:**
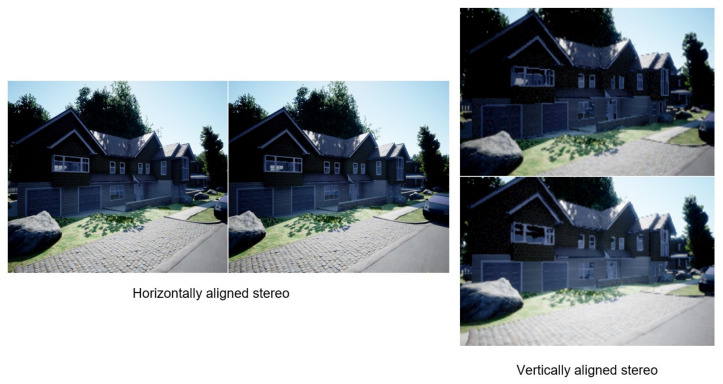
Two different possible configurations, horizontal stereo configuration and vertical stereo configuration obtained from AirSim.

**Figure 11 sensors-23-01134-f011:**
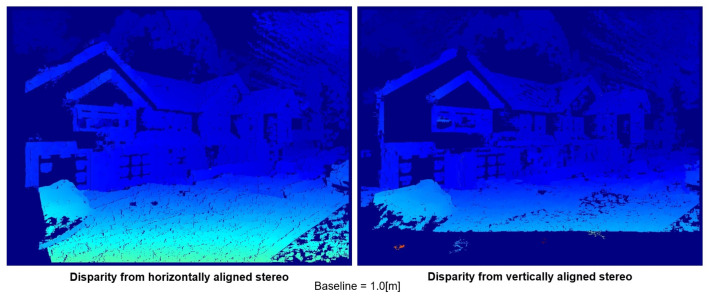
Comparison of locally normalized disparity images from horizontal stereo configuration (**left**) with vertical stereo configuration (**right**).

**Figure 12 sensors-23-01134-f012:**
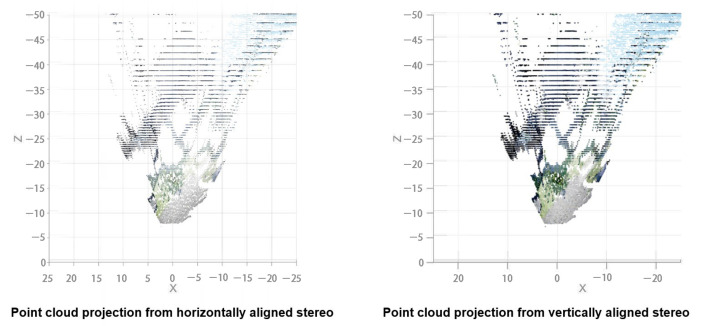
Comparison of point cloud projected from horizontal stereo configuration (**left**) with vertical stereo configuration (**right**). No major dissimilarity is observed in terms of range estimation.

**Figure 13 sensors-23-01134-f013:**
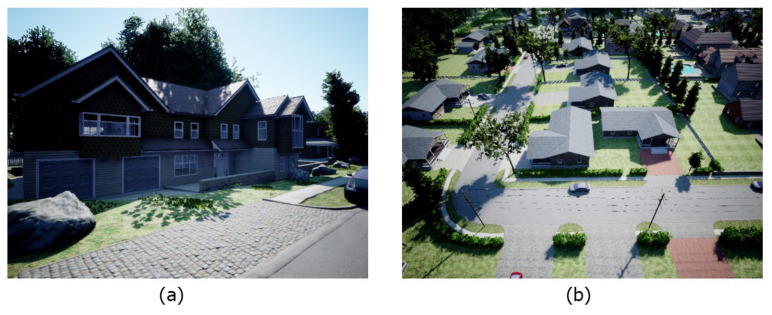
Picture of the two benchmark area of interest (**a**) and area of interest (**b**) obtained from AirSim.

**Figure 14 sensors-23-01134-f014:**
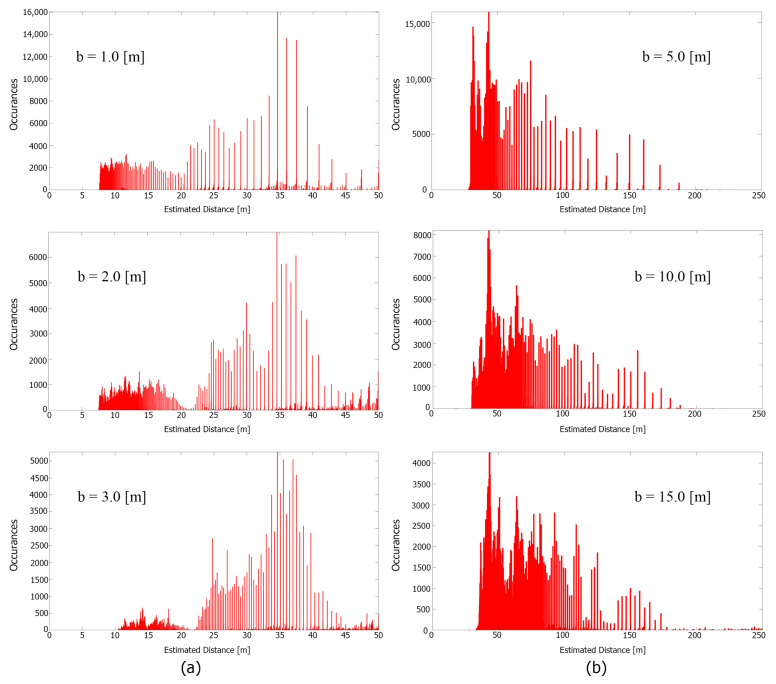
Depth estimation distribution graph with their corresponding baseline from area (**a**) (**left**) and area (**b**) (**right**), where the x-axis is the estimated distance in m, and the y-axis is the number of occurrences.

**Figure 15 sensors-23-01134-f015:**
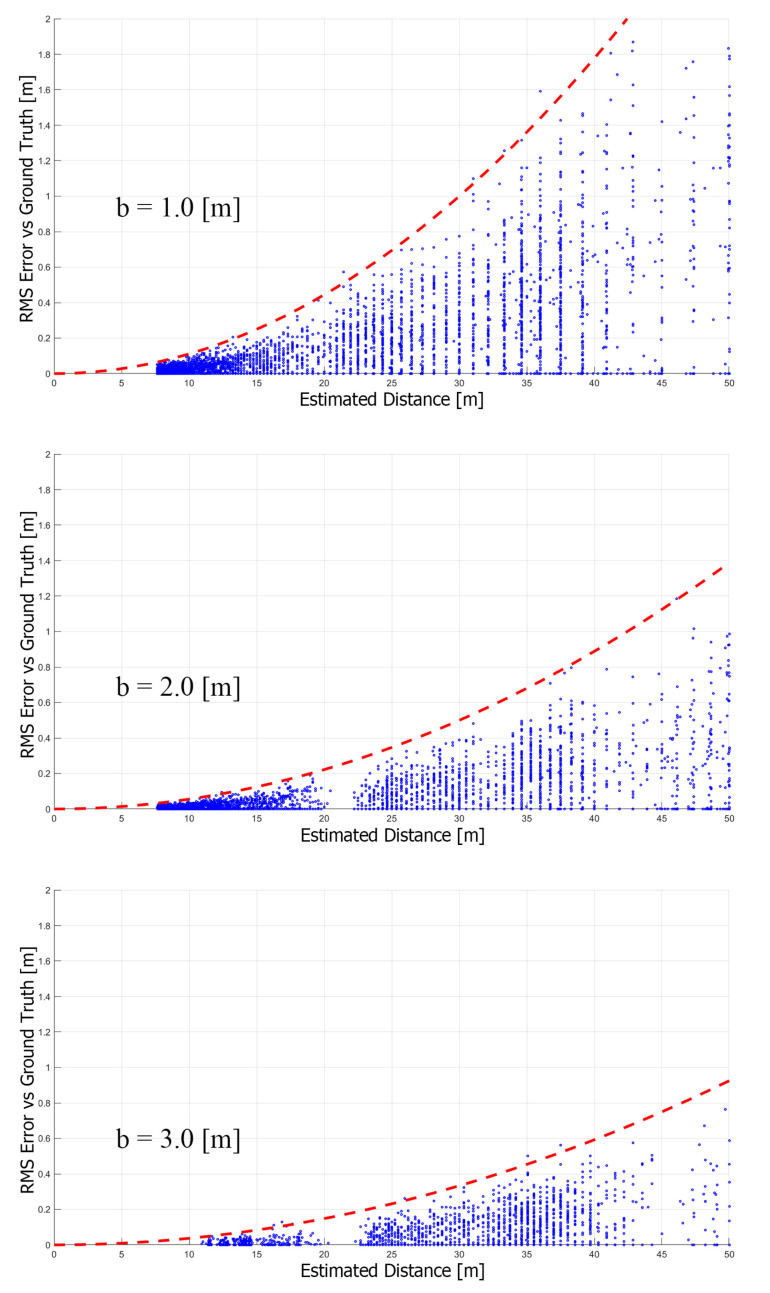
Error of estimated depth of area of interest (a) compared to ground truth in 1/100 density. The x-axis is the estimated depth, the y-axis is the estimation error, and the red dashed line is the estimated error given by the theoretical equation.

**Figure 16 sensors-23-01134-f016:**
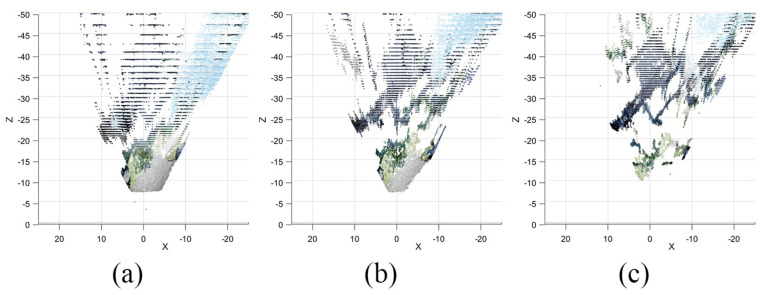
Point cloud from depth estimation of (**a**) baseline 1.0 m, (**b**) baseline 2.0 m, and (**c**) baseline 3.0 m seen from top–down view.

**Figure 17 sensors-23-01134-f017:**
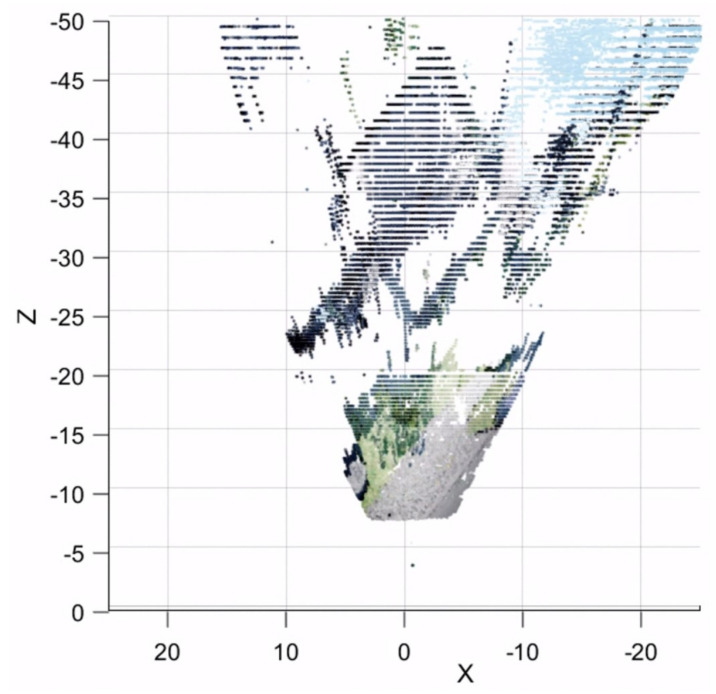
The resultant point cloud from baseline fusion of three baselines seen from top–down view.

**Figure 18 sensors-23-01134-f018:**
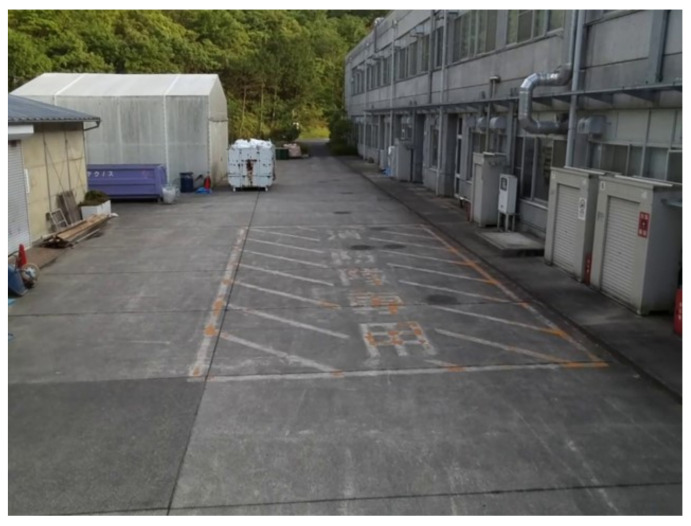
The testing area of interest of a parking area.

**Figure 19 sensors-23-01134-f019:**
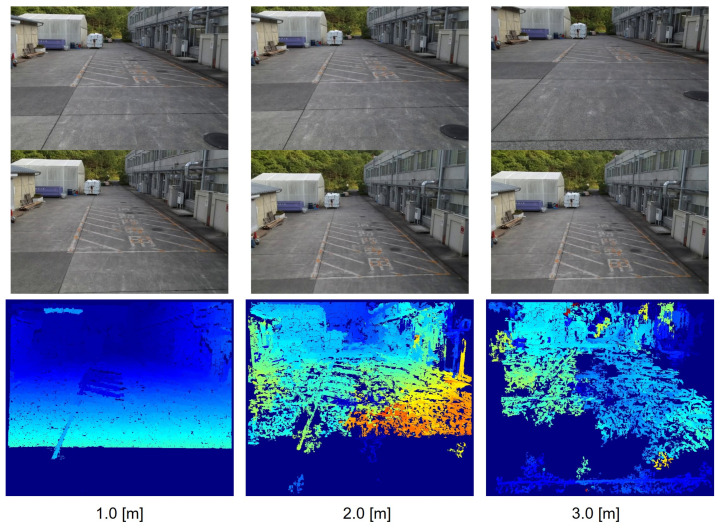
Two image frames and their disparity from each respective baseline.

**Figure 20 sensors-23-01134-f020:**
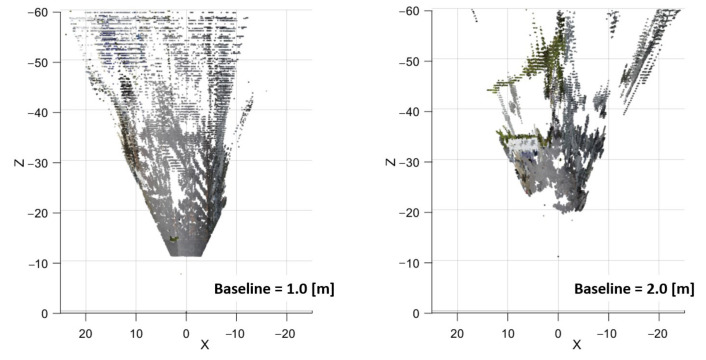
Point cloud generated from 1 and 2 m baseline seen from top–down view.

**Figure 21 sensors-23-01134-f021:**
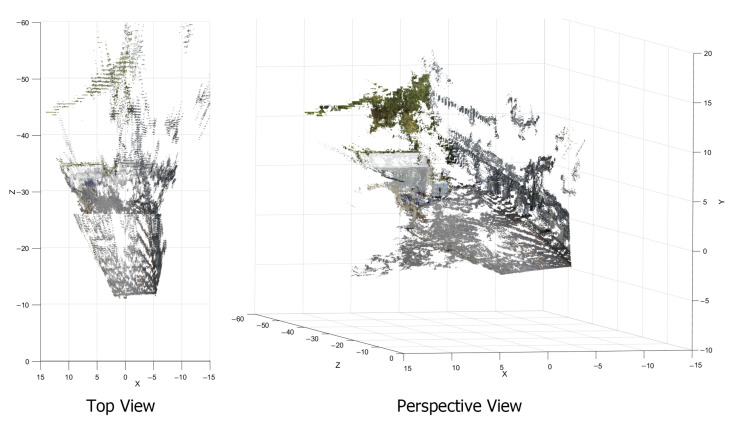
Resultant point cloud from baseline fusion.

**Figure 22 sensors-23-01134-f022:**
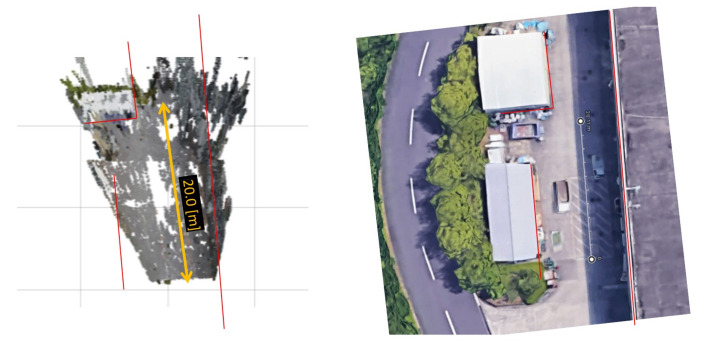
Shape and scale comparison of the resultant point cloud to ground truth obtained from Google Maps.

**Table 1 sensors-23-01134-t001:** (a) General characteristic of DJI Tello. (b) Characteristic of Optitrack V120 Duo motion capture device.

(a)		(b)	
Camera Resolution:	960 × 720 pixel	Resolution:	640 × 480 pixel (×2)
Dimensions:	98 × 92 × 41 mm	Dimensions:	41 × 279 × 51 mm
FOV:	82.5 deg	FOV (HxV):	47 × 43 deg
Weight:	80 g	Weight:	600 mm
Focal Length:	5 mm	Frame Rate:	120 fps
Payload (centered):	<80 g	IR Wavelength:	850 nm

## Data Availability

Not applicable.

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
