# Peer review of "Variable Baseline and Flexible Configuration Stereo Vision Using Two Aerial Robots"

_sensors, 2023, doi:10.3390/s23031134_

Round 1

Reviewer 1 Report

Excellent job, congratulations.

Reviewer 2 Report

This paper proposed a variable baseline and flexible configuration stereo setup using aerial robots. It is quite interesting that you tried to find an optimal multi-baseline stereo setup to maximize the effects of multi-baseline stereo and point cloud fusion. It would be better if we could find an optimal fusion algorithm more effectively considering the reliability of each point cloud point instead of selecting point cloud points mainly depending on the distances from the camera(depth). In addition, you mentioned 3D monitoring of an area in close-to-real time in the conclusion. It would be good to show running time comparison results between your algorithm and other conventional algorithms. Moreover, it would be better to show the performance of the point quantitatively of each baseline stereo setup and the proposed fusion algorithm. 

Reviewer 3 Report

The manuscript "Variable Baseline and Flexible Configuration Stereo Vision Based on Two Aerial Robots" proposes a new method for stereo vision remote sensing using aerial robots. The flexible configuration of stereo vision is achieved by separating the left camera and the right camera on two independent quadrotor aerial robots. This manuscript also proposes a practical application method of variable baseline stereo vision combining multiple stereo baseline point clouds to solve the problems caused by using inappropriate baselines. The manuscript is well structured and logical. There are still some minor issues that need to be addressed by the authors.

1. In line 47, this manuscript mentions that the advantage of this method is that the device takes up less resources and only needs one color camera. However, in practice, two monocular cameras and a binocular infrared camera for object tracking are used.

2. In line 130, the authors claim that this method does not require camera pre-calibration. I guess this is because of the existence of the IMU. The statement here is easily misunderstood because of the infrared camera.

3. In line 158, a colon is used, so the period in line 159 should be a semicolon.

4. In section 3.2 and section 3.3, I did not understand how the author used the latest offset to eliminate the cumulative error of the IMU. Since the cumulative error of the IMU is very large, if it cannot be eliminated well here, it will have a great impact on the accuracy of the 3D reconstruction.

5. In line 206, Equation (5), since the starting positions of the two drones are different, why can the same W be used to represent the world coordinate system, should the two coordinate systems be distinguished, or should be explained more clearly
